# A Feasibility Study of a Traffic Supervision System Based on 5G Communication

**DOI:** 10.3390/s22186798

**Published:** 2022-09-08

**Authors:** Allan Tengg, Michael Stolz, Joachim Hillebrand

**Affiliations:** 1Virtual Vehicle Research GmbH, 8010 Graz, Austria; 2Institute of Automation and Control, Graz University of Technology, 8010 Graz, Austria

**Keywords:** automated driving, 5G networks, traffic simulation, SUMO

## Abstract

At present, autonomous driving vehicles are designed in an ego-vehicle manner. The vehicles gather information from their on-board sensors, build an environment model from it and plan their movement based on this model. Mobile network connections are used for non-mission-critical tasks and maintenance only. In this paper, we propose a connected autonomous driving system, where self-driving vehicles exchange data with a so-called road supervisor. All vehicles under supervision provide their current position, velocity and other valuable data. Using the received information, the supervisor provides a recommended trajectory for every vehicle, coordinated with all other vehicles. Since the supervisor has a much better overview of the situation on the road, more elaborate decisions, compared to each individual autonomous vehicle planning for itself, are possible. Experiments show that our approach works efficiently and safely when running our road supervisor on top of a popular traffic simulator. Furthermore, we show the feasibility of offloading the trajectory planning task into the network when using ultra-low-latency 5G networks.

## 1. Introduction

Connected automated driving is currently a popular research topic. In recent years, a lot of effort has been put into the development of different car-to-car and car-to-infrastructure communication systems. Both local inter-vehicle communication such as WiFi 802.11p [1], as well as cellular 4G technologies [2] were evaluated for different scenarios. With the emergence of the 5G standard, it is now possible to unify both approaches within a common communication protocol. The promised ultra-low latencies, the high communication bandwidth and the enhanced communication reliability of 5G [3] seem to make it a good choice for connected automated driving.

Currently, the path- and trajectory-planning of automated vehicles [4] is solely performed based on local sensor data and map information. Considering driving velocities, and thus the required communication delays, this restriction on local sensor data is understandable. However, this determination also has many downsides: only objects within the vehicle’s field of view can be detected and taken into account. It would be desirable to have a better overview in order to make the right decisions, especially in traffic jam situations. Equally advantageous would be to know the exact intention of another road user as proposed by [5] and not have to guess it—as is the case in current implementations.

Given the promising characteristics of 5G, the widely used vehicle-centric approach has to be questioned. By exchanging location, speed, heading and other relevant data of automated vehicles using a low latency network, the trajectory planning could be improved. In our research project, we even go a step further and explore the possibility of outsourcing the trajectory planning to a powerful server of the 5G network operator. This way, the individual trajectory planning of a vehicle is replaced by an overall planning, covering all vehicles on a monitored road section. Similarly to a tower in air-traffic, the actions of the autonomous vehicles on the road are coordinated and consequently the flow of traffic is optimized. This way, the energy consumption could be lowered and traffic congestion reduced. However, unlike in airspace surveillance, not all vehicles are equipped yet with connectivity. This must be taken into account when designing a centralized trajectory planning strategy.

In this paper, we describe an algorithm that is capable of performing trajectory planning for several automated vehicles on a given road section. Based on the periodic reports from the autonomous vehicles under supervision, the algorithm plans and recommends trajectories for all of them. Provided that the recommendations are plausible, the connected automated vehicles are supposed to follow them. Conventional vehicles need to be taken into account as well, in the sense that they are detected by the on-board sensors of the autonomous vehicles and predicted. The feasibility of the approach is demonstrated using SUMO [6]—a microscopic and continuous multi-modal traffic simulation package. A sensitivity analysis is made by inserting artificial delay times into the communication between the vehicles and the trajectory planning running on a centralized server. This way, the requirements regarding communication delays can be identified.

The idea of connected automated vehicles and centralizing the trajectory planning is not new. Several activities can be found in the literature. For example, in [7] benefits of connected automated driving in terms of traffic capacity are demonstrated. A similar—yet more detailed approach—is presented in [8]. Optimizing urban traffic throughput using connected automated vehicles is discussed in [9]. Planning a trajectory for on-ramp merging is the focus of [10,11,12].

In contrast to these research activities in the domain of connected automated driving [13,14], we propose a generic algorithm which is not tailored to any specific traffic situation (e.g., intersection, on-ramp), but is applicable to a wide range of scenarios. Instead of finding a solution for a specific situation, we propose the following way: we assess and quantify a traffic situation, and try to optimize it by varying the degrees of freedom available in the scenario. Admittedly, this approach will not lead to the perfect solution, although in most cases it is not necessary. Considering the inaccuracies of sensor data and communication latency, a perfect solution in theory is very likely not feasible in practice anyway.

The remainder of this paper is structured as follows. Section 2 describes our concept of simultaneous trajectory planning for multiple vehicles. Section 3 gives the details of the implemented genetic optimization algorithm and applied heuristics. The interaction between the optimization algorithm and SUMO is described in Section 4. Section 5 summarizes the results of our latency sensitivity analysis, before Section 6 concludes the paper.

## 2. Problem Statement

In this chapter, we explain our approach of simultaneous multi-trajectory planning. Several assumptions and simplifications are made in order to keep the proposed methodology as elementary as possible.

Firstly, the entire paper uses the s–d coordinate system, where s is along the road and d is oriented orthogonal to the road. This implies that entry- or exit-lanes are not supported for now. Moreover, the number of lanes is assumed to be fixed for the section under consideration. It is also expected that traffic flow is in one direction only. With regard to road users, a distinction is made between connected automated vehicles (green), conventional vehicles (yellow) and static obstacles (gray) (cp. Figure 1).

Connected automated vehicles report their status and their perceived surroundings periodically, while conventional vehicles can be only detected indirectly. Road construction sites and lane reductions can be modeled by placing large static obstacles on the road. On the actuator side, it is assumed that every connected automated vehicle is capable of following a lane using a given speed profile, and is able to trigger a lane change upon request. For safety reasons, each connected automated vehicle must verify all received directives for plausibility before execution.

Every planning cycle starts on the basis of the reports of the connected automated vehicles. Their precise location, speed and acceleration is put in the model of the road. All conventional vehicles picked up by the onboard sensors are also added to the model. While the position and velocity of the conventional vehicles can be determined fairly directly with modern sensors (e.g., Lidar, Radar), this is not the case with acceleration. Therefore, acceleration is not available for conventional vehicles. Finally, the predefined static obstacles are also placed in the model.

The challenge is now to determine an optimal trajectory for every vehicle, including the required speed profile and indicating whether/when a lane change is necessary. These trajectories must of course be free of collisions. However, there are several additional requirements for these trajectories, such as a safety margin which must be maintained around every vehicle. Furthermore, traffic rules, such as speed limits or right hand driving, must also be considered. Vehicles with higher priority (e.g., an emergency vehicle) should not be hindered. In general, strong braking and acceleration maneuvers should be avoided to reduce energy consumption and maintain passenger comfort.

As all vehicles share the same road, the actions of one vehicle potentially influence the behavior of all surrounding vehicles. Especially in dense traffic, an optimization of individual vehicle trajectories is thus not advised. Therefore, we elaborated an optimization method—based on a genetic algorithm—that finds a set of trajectories for all vehicles under control simultaneously.

## 3. Genetic Algorithm

Of all optimization algorithms found in the literature, a genetic algorithm [15] seems to be most adequate for the task at hand. A genetic algorithm finds increasingly better solutions over time. In the real application, the runtime is strictly limited by the selected communication schedule. By tailoring the genetic algorithm to the specific problem, and by applying heuristics, it is usually possible to improve the convergence speed.

To solve any problem using a genetic algorithm, it is necessary to define a relation between the basic elements of the algorithm (gene, chromosome, population), and the actual optimization problem. Furthermore, the two main operations of the genetic algorithm (mutation and crossover) must be implemented accordingly. For evaluating and comparing potential solutions, a fitness score must be calculated—the higher this value is, the better the solution is. On the basis of these operations, the optimization then runs analogously to the evolution of biological life: starting from an initial population of approximately 50 individuals, their fitness is determined, the fittest are selected for reproduction, and the next generation is formed by the mutation and recombination of the fittest individuals.

### 3.1. Mapping of Genes

The trajectory planning problem can be mapped to a gene representation (cp. Figure 2) in a meaningful way: Every gene encodes the trajectory of a connected automated vehicle as tuple, consisting of the vehicle acceleration *a*, the direction $d_{ch}$ and the time $t_{ch}$ of the planned lane change. The acceleration is in the range between $-a_{min}$ and $a_{max}$, depending on the vehicle capabilities. Using the acceleration *a* instead of the velocity to describe the speed profile has the advantage that the resulting speed is always continuous, and a vehicle controller usually expects a torque as input. The lane change direction field $d_{ch}$ is limited to three possibilities: staying on the current lane; changing to the left lane; changing to the right lane. The time for the lane change $t_{ch}$ can be somewhere in the range of 0.0 s up to planning horizon (currently 7.0 s).

A chromosome consists of a sequence of *n* genes, where *n* equals the number of connected autonomous vehicles in the scenario. For an important heuristic presented later, it is necessary that the sequence of connected automated vehicles always remains the same throughout successive optimization runs. The mutation operation applied to a gene is implemented as the randomization of one of its elements (*a*, $d_{ch}$ or $t_{ch}$) within the possible range. The crossover operator is implemented as an N-point crossover according to literature.

### 3.2. Definition of Fitness Function

As mentioned above, the fitness function is used to evaluate a chromosome/solution. Generally speaking, a decent fitness function should be strictly monotonous to guide the optimization toward good solutions. Likewise, a better solution should be awarded a higher fitness score; a solution with many violations should be punished more than a solution with a single violation.

Several aspects provide a contribution to the fitness score calculation: The more distance an individual vehicle drives in the duration of the planning horizon, the better. On the other hand, driving too close to another vehicle causes reductions in fitness score. If a collision is detected, the fitness needs to be significantly decreased. A design decision has been made that a negative fitness value should correspond to an invalid trajectory. Based on the allocated priority, it is possible to grant certain vehicles the right of way—emergency vehicles for instance.

To actually calculate a fitness score for a given potential solution, it is necessary to perform a micro simulation: Initially, all vehicles are placed at their starting location. Then, they are incrementally moved by Δt=100 ms according to their motion parameters until the planning horizon ($T_{lookahead}=7\;$ s) is reached. The distance traveled during this time is accumulated. After every simulation step, the gaps between all adjacent vehicles are checked and the collisions are counted. Based on this micro simulation, a partial score is calculated for every connected automated vehicle. In case no collision occurred, the following equation is used:$$score_{car}=min\left[\Delta s-\frac{\Delta s\cdot \left(t_{dist}-T_{begin}\right)}{T_{gap}-T_{begin}},\Delta s\right]\cdot {p_{lchg}}^{m}$$

In the above equation, Δs denotes the traveled distance of a connected automated vehicle during the entire planning horizon ($T_{lookahead}$). The temporal distance between the connected automated vehicle and the vehicle ahead is given by $t_{dist}$. As long as the vehicle in front is far ahead ($T_{begin}$), no punishment is subtracted. However, if the vehicle in front gets closer than $T_{begin}$, the punishment is executed. In the case of the vehicle in front being closer than $T_{gap}$, the score turns negative, indicating an invalid trajectory.

In case a collision is detected during the planning period, a different equation is used to calculate the fitness:$$score_{car}=\Delta s-\frac{\Delta s\cdot \left(t_{col}-T_{lookahead}\right)}{T_{avoid}-T_{lookahead}}$$

Depending on the time of the collision $t_{col}$, a punishment value is subtracted from the distance traveled Δs during the planning horizon. The expression differentiates between a collision that can be avoided (later than $T_{avoid}$) and a collision that is inevitable, resulting in a negative fitness score. A graphic representation of these parameters is depicted in Figure 3.

The combination of all individual scores is delicate. If only one vehicle violates a safety gap, the entire fitness score must become invalid. For this purpose, a theoretical maximum score is determined, assuming that every vehicle moves with its maximum velocity:$$score{}_{max}=\sum_{all\;cars}prio_{car}\cdot v_{max_{car}}\cdot T_{lookahead}$$

Overall fitness score of the chromosome is then calculated using the following expression:$$fit=\sum_{all\;cars}\left(score_{car}\cdot prio{}_{car}\right)-\#vio\cdot score{}_{max}$$

Every individual score is summed up and weighted by the vehicles’ priority. For every detected violation of a safety gap, $score{}_{max}$ is subtracted as punishment. This way, a negative overall fitness score is guaranteed whenever a single violation is detected. More violations cause a lower score. In case no violations are detected, the distance traveled and the gradual punishments for driving too close drives the optimizer toward the optimal solution.

### 3.3. Improved Optimization Loop

Figure 4 shows the genetic algorithm main loop (blue) embedded in the data exchange process with the connected automated vehicles (green). Every simulation step starts by getting the position, speed and other values for both the connected automated vehicles and the conventional vehicles sensed by the automated vehicles. This is the starting point for any further planning and extrapolation. To start with, an initial population of solutions is randomly created. Unlike the genetic algorithm found in the literature, the initial population also includes the solution from the last time step, after shifting the time of the lane change by ΔT=100 ms. This improves the continuity of planning. On top of that, it massively accelerates the convergence, because in many cases, a good solution is probably still good 100 ms later.

Then, a fitness score is calculated for all potential solutions using the equations shown above. Then, the selection for reproduction follows. Solutions with higher fitness scores are more likely to reproduce using the mutation- or crossover-operation. Elitism—the best solution which is always promoted into the next generation—is enabled to improve continuity. New solutions created by these two operations form the next generation of chromosomes and the algorithm starts again with the calculation of the fitness score.

The optimization loop continues until the time for optimization has passed. If the best solution found so far is valid (fit>0), then it is immediately sent as a recommendation to the connected automated vehicles. However, if the best solution is invalid, a chromosome repair function is triggered. This might be caused by a conventional vehicle changing the lane unexpectedly with short notice. By issuing a strong braking command to the appropriate connected autonomous vehicle, the chromosome repair function manages to avoid the accident. Needless to say, passenger comfort and energy efficiency are secondary considerations in such a rare intervention.

### 3.4. Heuristics for Increased Performance

Due to the strictly limited time for the optimization, both heuristics to accelerate the convergence and methods to narrow down the search space are desirable. Regarding the acceleration of the convergence, the promotion of applied trajectories in the next simulation step—as explained in paragraph Section 3.3—is already a major improvement. By discretizing the values in the chromosome while suppressing duplicates within the population, the search space can be narrowed. In the present implementation, the times of lane changes are discretized to a multiple of 100 ms, and the acceleration/deceleration is normalized to ±100% with a step size of 1%.

## 4. Implementation with SUMO

Designing, implementing, optimizing and testing such an approach on a fleet of real automated connected vehicles would be extremely time consuming and dangerous. Instead, an appropriate simulation environment had to be used. From the variety of vehicle- and traffic-simulation tools available nowadays, SUMO [6] was chosen. Among other advantageous features (i.e., freely available), the level of abstraction, as well as the open interface *TraCI*, were essential for this decision. This way, the implementation could be tested and the parameters optimized for a variety of traffic situations.

### 4.1. Scenario for Sumo Simulations

Due to the integration of this research topic into the EU project *5G-Carmen*, a section of the Brenner motorway in Austria had to be used as a virtual test track. The scenarios to be investigated were also influenced because of that.

In the context of this paper, an emergency vehicle scenario is replicated in the SUMO simulator. This particular scenario revolves around an emergency vehicle, which is on duty on a busy motorway. By using connected automated vehicles and an appropriate supervisor algorithm, it should be demonstrated that the approaching emergency vehicle is given the right of passage by other connected automated vehicles.

More precisely, a 3 km-long motorway section of the Austrian A13 between the exit Gries West and Brennersee was modeled within SUMO. To increase the solution domain complexity, the virtual test track has three lanes instead of the actual two lanes. The right-hand drive requirement is turned off in the simulation for further flexibility.

On this motorway, a basic traffic jam consisting of cars and trucks was generated randomly by SUMO. After the start-up phase, there are on average 50 conventional vehicles on this motorway section. Then, 10 additional connected automated vehicles are spawned at the motorway on-ramp with a 3 s gap in between. The last spawned connected automated vehicle is the emergency vehicle. It is basically the same as the other nine CAVs before but it has been assigned a higher priority. Once a connected automated vehicle merged into the traffic on the motorway, the supervisor takes control. The simulation is completed as soon as the emergency vehicle reaches the exit Brennersee. In case the emergency vehicle collides with another road user, the simulation is prematurely aborted.

### 4.2. Interaction with Sumo

The interaction between the traffic supervisor and SUMO is depicted in Figure 5. The SUMO traffic simulation at the bottom contains both, the connected automated vehicles and the base traffic, comprised of conventional vehicles. Every connected automated vehicle within SUMO is linked to an agent using the TraCI interface. The agent is basically an object written in C++ that interacts with the Road Supervisor. By using TraCI, the flow of time in the simulation can be precisely controlled. For the sake of simplicity and performance, all communication delays, determined by the Simulated Network, are converted into position shifts of the corresponding vehicles, based on their current velocity and acceleration. In return, the time interval of the simulation remains constant (100 ms), and all TraCI queries can be done simultaneously in a batch instead of many individual queries to SUMO, reducing the communication overhead. Moreover, the job of the agents is to query position, speed, acceleration, driving lane and the surrounding vehicles of the associated SUMO vehicle. For a more realistic behavior, the vehicles’ sensor range is assumed to be limited to −100 …200 m around the vehicle. Lane changes and target speed requests need to be transferred to SUMO, respectively.

In addition to communication aspects, testing the traffic supervisor using SUMO also has implications on the optimizer and its parameters. When using the default SUMO configuration, every vehicle under SUMO control strictly maintains a distance from the vehicle ahead of at least 1 s. If, for any reason, this constraint is violated, SUMO initiates an emergency braking of the affected vehicle. Thus, the parameter $T_{gap}$ needs to be set to 1.5 s in the optimizer to preserve a 0.5 s margin. The associated parameter $T_{begin}$ is set to 3 s, guiding the optimizer towards platooning with the temporal separation of 3 s between individual vehicles.

### 4.3. Configuration of a Scenario

To create a scenario, it is necessary to define a set of connected autonomous vehicles with parameters. For this purpose, all automated vehicles in the scenario are described using a XML file (cp. Listing 1). Apart from obvious parameters such as length, width and color, there are also some parameters that require further explanation: maxAccel defines the maximum acceleration that a connected automated vehicle is capable of, regarding its engine power and vehicle mass. Currently, this value is assumed constant, but for a future real-world implementation, it would be possible to update this value periodically based on velocity, road slope, engine temperature and battery charge. The parameters startLane and Offset are forwarded to SUMO, selecting the lane and the lateral offset on the lane where the vehicle must be spawned. Likewise, Route is forwarded to SUMO, indicating the sequence of roads that the new vehicle should follow. startTime is the delay (in seconds) that should be waited before the vehicle is actually created. This is needed to wait for the base traffic to fill the road and define intervals between several connected automated vehicles. startSpeed defines the initial speed (in km/h) of the newly created vehicle.

The base traffic is generated the SUMO way by defining vehicle types and their probability of occurrence within the *rou.xml* configuration file. Several scenarios with a similar traffic density can be generated by starting SUMO with different seed values for the internal random number generator. Figure 6 depicts a running SUMO simulation consisting of several automated connected vehicles (green, red) and the base traffic (yellow).

**Listing 1.** Definition of a connected automated vehicle in XML format.
   <Vehicle Name="Autonom1" Type="normal_car">

      <Length> 4.0 </Length>

      <Width> 1.9 </Width>

      <maxSpeed> 130 </maxSpeed>

      <maxAccel> 2.0 </maxAccel>

      <startLane> 0 </startLane>

      <startTime> 30.0 </startTime>

      <startSpeed> 40.0 </startSpeed>

      <Route> myroute </Route>

      <Offset> 5.0 </Offset>

      <Color> #00ff00 </Color>

   </Vehicle>


### 4.4. Assumptions and Limitations of Implementation

Since the focus of this work is set on the feasibility of this method and the influence of communication delays, many details have been neglected. Converting real-world longitude- and latitude-reports into s/d coordinates is probably the most important simplification. For this to be done on a motorway section, a HD-map must be available and used. Sensor synchronization among the vehicles using GPS is another neglected job to avoid sensor jitter on top of communication jitter. Moreover, the entire automated driving functionality within the vehicle is disregarded in this work. The recommendations from the road supervisor should be verified and implemented there. In case of conflicting recommendations, the vehicle should rely on its onboard sensors and take actions to avoid collisions. The vehicles must of course be able to drive fully autonomously in the case of communication loss. This method is not meant to be a replacement of the vehicle’s intelligence but a guidance aid for congestion-prone sections of roads.

## 5. Results

This chapter demonstrates two aspects. Firstly, the basic function of the presented algorithm and secondly, the effect of communication delays.

Considering the way that the acceleration values are determined by the genetic optimization algorithm one might be concerned about the driving comfort. Therefore, the speed profile of a high-priority emergency vehicle during a simulation run is plotted in Figure 7. Overall it looks very reasonable. In sections where the emergency vehicle had to follow other vehicles, the speed is jittery, but the oscillations are within small margins. Towards the end of the simulation, the vehicle reaches the speed limit of 130 km/h and is thus limited.

Almost as in a chaotic system, minor changes in the input may lead to completely different results. The reason is the highly interactive behavior with many cause-and-effect loops between the traffic participants. The actions of one vehicle may change the behavior of other vehicles in the simulation, and these changes may feed back to the initial vehicle. This makes the presentation of results and the comparison with other approaches quite challenging. Instead of comparing a single simulation run, a bunch of simulations with different seed values are needed to determine a trend.

After every simulation run, a statistic is reported. This includes the time it took the emergency vehicle to reach its goal, and the number of collisions that occurred during the simulation. Furthermore, the strong and emergency braking maneuvers of all vehicles are counted. Regarding a communication latency sensitivity analysis, the communication delay range is recorded. For debugging purposes, the seed values are stored as well.

It is assumed that the supervisor broadcasts a query to all vehicles under supervision. Because of network communication latencies, this query arrives more or less delayed at the vehicles. The effect of these random delays can be seen in Table 1. There, the same seed has been used for every simulation but the communication latency has varied between 0 ms and 220 ms (first column). The number of significant braking maneuvers of all cars within the simulation are counted in the second column. The threshold for a significant deceleration is currently set to 1 m/s^2^. It is assumed that electric vehicles can recuperate the braking energy up to this deceleration value with high efficiency. Even stronger braking interventions of 4.5 m/s^2^ and more are assumed to be instances of emergency braking. These are extremely inconvenient for the passengers and should be avoided. The next column counts the collisions that occurred during the simulation run. The duration of the simulation is listed in the last column of Table 1. In the presented scenario, the simulation finishes as soon as the emergency vehicle reaches the exit lane or is involved in a crash. If the simulation terminates prematurely (e.g., latency = 190 ms), the values are not representative and therefore cannot be used for further evaluation.

As can be expected, with increasing communication latency, significant brakings and emergency interventions increase. With very high communication latencies, collisions start to occur. By varying the base traffic seed value while keeping the traffic density similar, the plots in Figure 8 and Figure 9 have been created. The results of 10 scenarios are plotted simultaneously, and a trend line was fitted. The box-plot has three components: (i) a blue box spanning the interquartile range (IQR) with a green point marking the median; (ii) a thin error-bar whose whiskers span 1.5 * IQR; and (iii) additional blue points marking outliers, that are data outside the whiskers.

It can be seen that the number of significant braking interventions (cp. Figure 8) increases quickly with increasing communication latency. Transferred to the real world, this means a deterioration in energy efficiency. In contrast, the number of emergency braking maneuvers remains low up to 20 ms communication latency, and starts to increase significantly after 30 ms.

### Requirements on Communication Layer

Regarding the deployment of such an approach using 5G networks in the future, requirements regarding the communication layer are necessary. Depending on the actual coding, the messages between supervisor and vehicle can be expected to be less than 100 bytes each, and are transmitted 10 times per second. With the high bandwidth of 5G networks, this transmission should not be an issue. More demanding is the application in terms of communication delays. Derived from Figure 8, it can be concluded that even short communication delays noticeably degrade the performance. According to Figure 9, the communication delays should definitely not exceed 20 ms in order to apply the presented algorithm to a real-world scenario.

## 6. Conclusions

In this paper, we presented a novel approach for a multi-vehicle path planning algorithm that can be used in congestion-prone road sections for automated connected driving solutions. By delegating the decision of target speed selection and lane changes to a centralized supervisor, foresighted planning can be achieved. Since the solution space grows quickly with the increasing vehicle count, a brute force optimization approach is not feasible. Therefore, a genetic optimization algorithm was developed that gradually improves the trajectories of the automated vehicles. By connecting several optimization runs over time, the faster convergence and a temporal continuity of the solutions is accomplished.

Our analysis confirms that 5G perfectly fits this kind of application, since it is capable of providing deterministic communication with latencies below 10 ms [16]. In contrast, 4G communication delays in a typical round-trip latency range of 40 ms and above is not adequate to implement the described approach in a meaningful way on a highway.

Currently, the presented algorithm is only operational in a simulation environment. Subsequently, the aim was to implement this approach on our autonomous prototype vehicles and evaluate it under real driving conditions—initially on test tracks, and subsequently on public highways. Thinking even further into the future, the algorithm would have to be optimized to work with a reasonable number of vehicles. Hence, the choice of the genetic algorithm, due to the critical part—the fitness calculation—can be easily parallelized and scaled up to realistic sizes. The ultimate goal would be to migrate the outlined approach to the infrastructure of the mobile network operator, as a so-called MEC service [17].

## Figures and Tables

**Figure 1 sensors-22-06798-f001:**
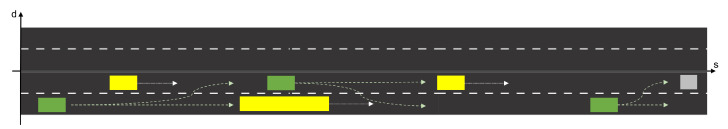
Connected autonomous vehicles (green), conventional vehicles (yellow) and static obstacles (grey) on a road model.

**Figure 2 sensors-22-06798-f002:**
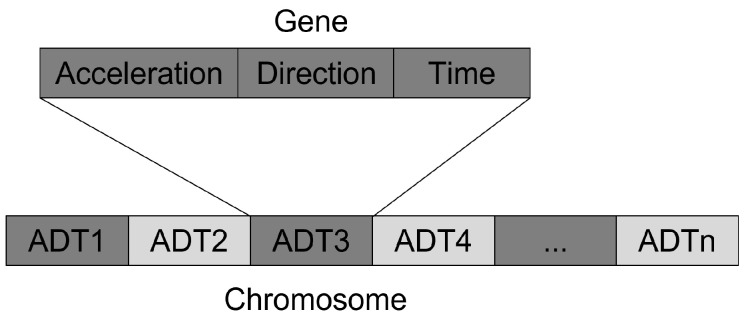
The mapping of the trajectory planning problem to a gene and chromosome representation for *n* connected automated vehicles.

**Figure 3 sensors-22-06798-f003:**
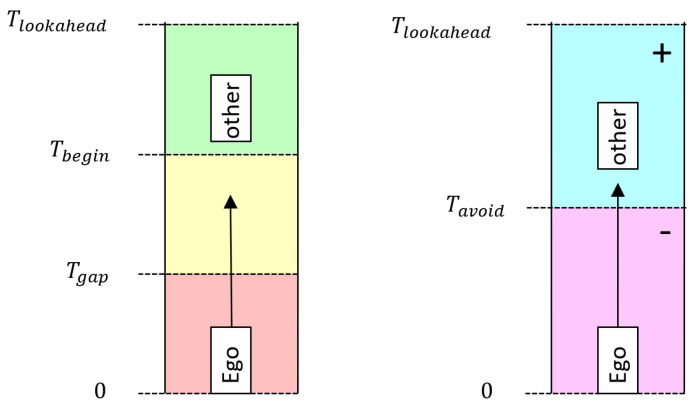
Graphic representation of $T_{gap}$, $T_{begin}$ and $T_{avoid}$.

**Figure 4 sensors-22-06798-f004:**
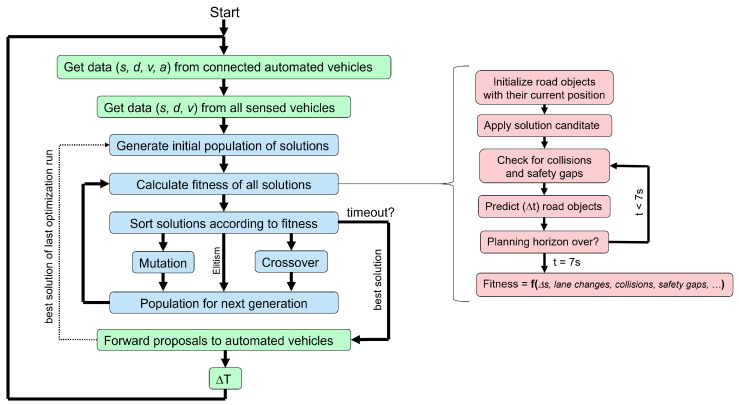
The genetic algorithm optimization loop including the micro simulation with granularity Δt for the fitness calculation, and the data exchange with the connected automated vehicles, using a communication interval of ΔT.

**Figure 5 sensors-22-06798-f005:**
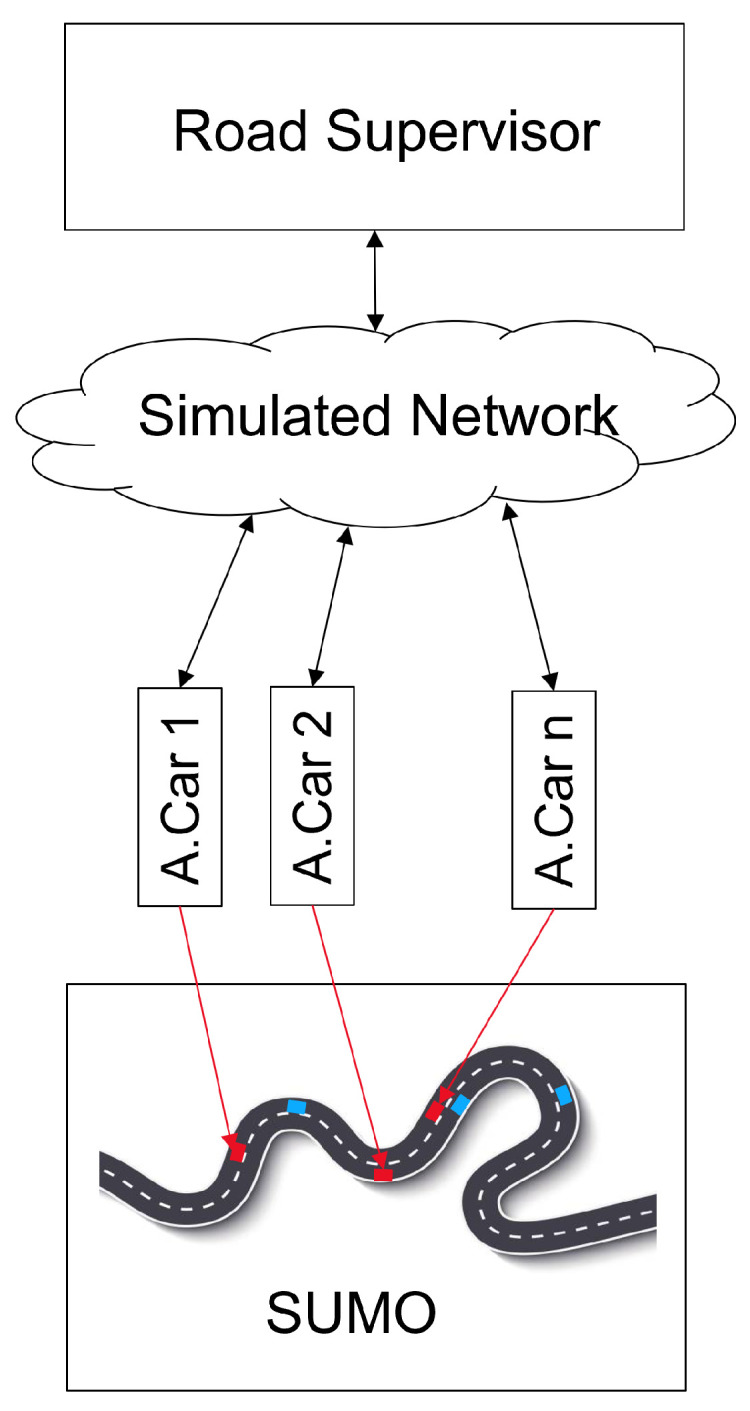
Road supervisor and SUMO connected via a simulated 5G network.

**Figure 6 sensors-22-06798-f006:**
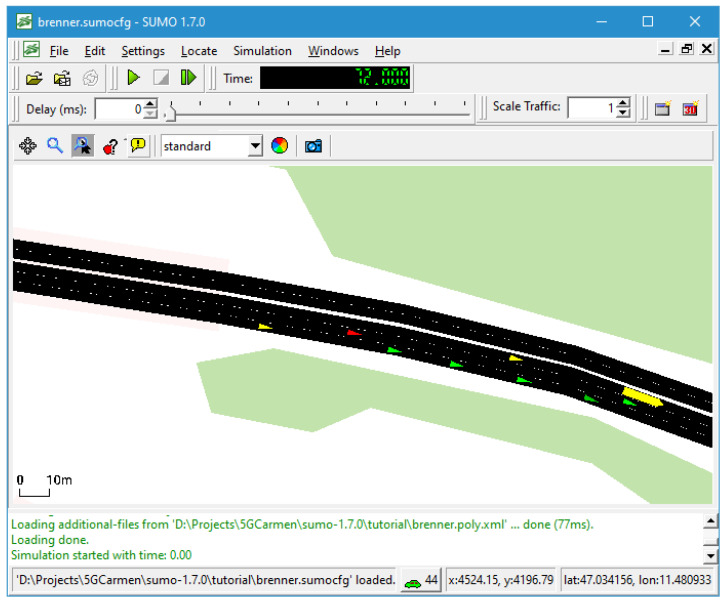
Screenshot of a SUMO simulation. Within the SUMO-generated base traffic (yellow), the connected automated emergency vehicle (red) overtakes normal connected automated vehicles (green).

**Figure 7 sensors-22-06798-f007:**
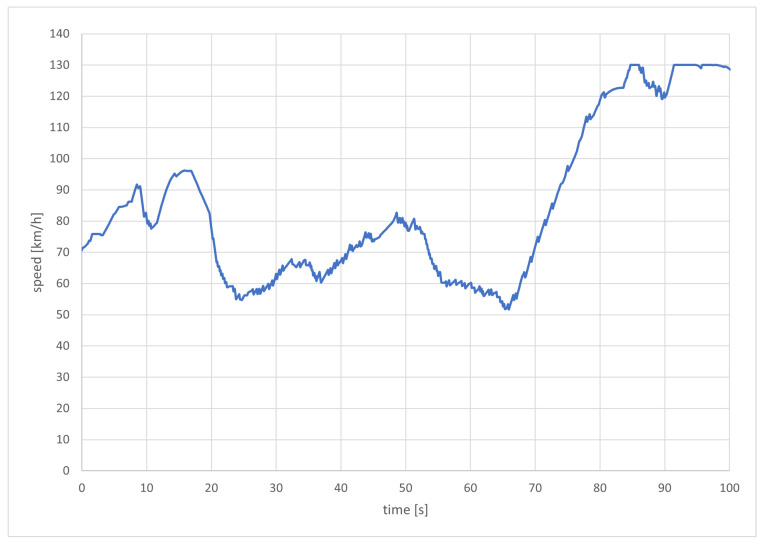
The speed profile of an emergency vehicle pushing through dense traffic.

**Figure 8 sensors-22-06798-f008:**
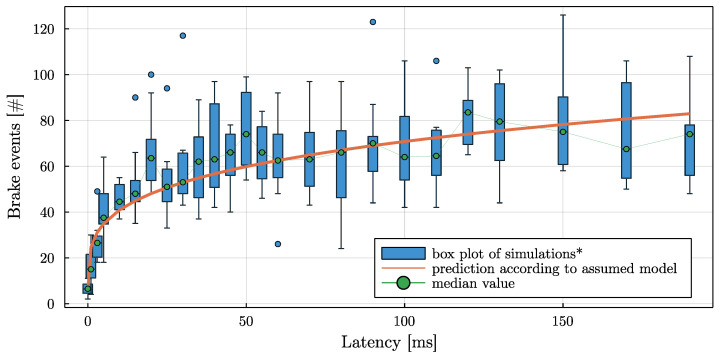
The number of significant braking requests with increasing communication latency between 0 and 220 ms over 10 scenarios.

**Figure 9 sensors-22-06798-f009:**
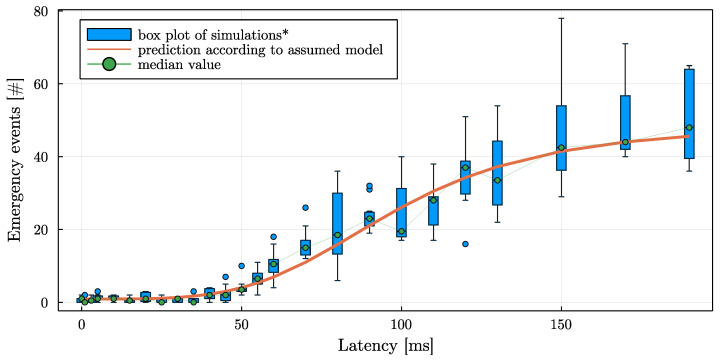
Emergency braking interventions when increasing communication latency between 0 and 220 ms.

**Table 1 sensors-22-06798-t001:** Result of a series of simulations using the same base traffic but different communication latencies.

Latency (ms)	Strong br. #	Emergency br. #	Collision #	Duration (s)
0	2	0	0	147
3	17	0	0	151
5	27	1	0	137
10	53	2	0	153
15	52	1	0	134
20	53	1	0	134
25	72	1	0	158
30	44	1	0	141
35	117	0	0	189
40	72	0	0	149
45	91	1	0	160
50	78	3	0	153
55	56	3	0	138
60	80	8	0	166
70	68	12	0	141
80	71	21	0	139
90	90	36	0	146
100	70	32	0	151
110	106	40	0	172
120	75	38	0	148
130	77	39	0	147
150	61	42	1	135
170	75	54	0	131
190	-	-	3	60 *
220	54	42	4	137

## Data Availability

Not applicable.

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
