# Peer review of "A Feasibility Study of a Traffic Supervision System Based on 5G Communication"

_sensors, 2022, doi:10.3390/s22186798_

Round 1
Reviewer 1 Report
A novel approach was proposed for a multi-vehicle path planning algorithm that can be used in congestion-prone road sections for automated connected driving solutions. The overall research idea of the article is clear, and the simulation results are reasonable. There are several issues to be clarified:
(1) How to ensure the accuracy of the simulation model? How to further improve the accuracy of the simulation model?
(2) How to verify the proposed algorithm with real vehicles (multiple vehicles) through 5G communication technology?
Author Response
(1) How to ensure the accuracy of the simulation model? How to further improve the accuracy of the simulation model?
We are using SUMO - a proven open source traffic simulation tool - to handle the simulation of all vehicles on the road. Using a plugin mechanism called "TraCI", the described approach is implemented into this simulator influencing particular vehicles in the simulation. Basically the speed profile and lane changes are triggered by the plugin according to the optimizatin presented in the paper. But accuracy and collision checks and other aspects of the simulation is taken care by SUMO.
(2) How to verify the proposed algorithm with real vehicles (multiple vehicles) through 5G communication technology?
Of course this is not a trivial task. We're planning to do it step by step. The first step is improving one of our three existing automated driving demonstrator vehicles to provide the interface and infrastructure needed (take speed- and lane change requests from the supervisor, output status information and objectlist of surroundings to supervisor). The second step is a rollout to all three vehicles. To further increase the fleet, there are plans to include manual driven cars and equip those cars temporarily with sensors like lidar or radar and high precision GPS and a screen to instruct the test driver to do. Besides that we're active in EU projects where connected driving is a topic, to find partners interested and to promote the idea and get access to their vehicles.
Reviewer 2 Report
Congratulations for the article written and for the field approached. Significant results and discussion are exposed however, some minor remark must be mentioned for improving the paper.
Line 79: How many lanes is considered in simulation in paper (different number of lanes in figures 1 and 6)? What is the length of simulated road section? What is the ratio between CAV and conventional vehicles?
Emergency vehicles noted in paper are considered as autonomous vehicles or conventional vehicles? How emergency vehicles are detected by algorithm?
How Tbegin and Tgap are determined?
Listing 1: Explain maxAccel value in comparison to values in lines 286-287.
Line 286-288: Can you better explain the difference between significant, strong and emergency braking which is confusing between text and table 1 and also brake events and emergency brake events in figures 8, 9.
Table 1: Is duration 60 s for latency 190 ms correct? Why there are missing values of strong br. and emergency br.?
Figure 8,9: Blue dots are not explained.
Author Response
Thank you for your very detailed review. Most of your comments have been integrated in the new version of the document and below there are just links to the changes, highlighted in blue in the revised version of the document.
Line 79: How many lanes is considered in simulation in paper (different number of lanes in figures 1 and 6)? What is the length of simulated road section? What is the ratio between CAV and conventional vehicles?
Figure 1 is just for the explanation of the problem. Figure 6 is the actual simuation. 3 lanes and 3 km road section. 10 vehicles are connected and approximately 50 vehicles are on this road section during simulation. The newly introduced Chapter 4.1 in the document explains more details.
Emergency vehicles noted in paper are considered as autonomous vehicles or conventional vehicles? How emergency vehicles are detected by algorithm?
Emergency vehicles report a higher priority. More details can be found in Chapter 4.1.
How Tbegin and Tgap are determined?
Tgap defines the minimum gap between two vehicles. It must be harmonized with SUMO, which uses as default 1.0s. We used 1.5s to have some margin. See chapter 4.2 for more details.
Listing 1: Explain maxAccel value in comparison to values in lines 286-287.
A short explaination of every parameter is now included in chapter 4.3.
Line 286-288: Can you better explain the difference between significant, strong and emergency braking which is confusing between text and table 1 and also brake events and emergency brake events in figures 8, 9.
Significant braking is a threshold where almost 100% recuperation of energy in an EV is no longer possible. Currently set to -1m/s². Emergency braking is a decelleration below -4.5m/s². This threshold is taken from SUMO, which reports strong brakings as warnings during the simulation. Chapter 5 has been extended to explain these thresholds.
Table 1: Is duration 60 s for latency 190 ms correct? Why there are missing values of strong br. and emergency br.?
The newly introduced chapter 4.1 explains the emergency vehicle scenario in detail. Basically, the simulation ends when the emergency vehicle reaches its goal or is involved in a crash. For 190 ms latency a crash happened and the simulation was terminated prematurely. Thus the results are incomplete.
Figure 8,9: Blue dots are not explained.
The plots have been updated and are explained within chapter 5.